# Distinct neural networks for the volitional control of vocal and manual actions in the monkey homologue of Broca's area

Natalja Gavrilov, Andreas Nieder*

Animal Physiology, Institute of Neurobiology, University of Tübingen, Tübingen, Germany

**Abstract** The ventrolateral frontal lobe (Broca's area) of the human brain is crucial in speech production. In macaques, neurons in the ventrolateral prefrontal cortex, the suggested monkey homologue of Broca's area, signal the volitional initiation of vocalizations. We explored whether this brain area became specialized for vocal initiation during primate evolution and trained macaques to alternate between a vocal and manual action in response to arbitrary cues. During task performance, single neurons recorded from the ventrolateral prefrontal cortex and the rostroventral premotor cortex of the inferior frontal cortex predominantly signaled the impending vocal or, to a lesser extent, manual action, but not both. Neuronal activity was specific for volitional action plans and differed during spontaneous movement preparations. This implies that the primate inferior frontal cortex controls the initiation of volitional utterances via a dedicated network of vocal selective neurons that might have been exploited during the evolution of Broca's area.

*For correspondence:
andreas.nieder@uni-tuebingen.de

**Competing interests:** The authors declare that no competing interests exist.

## Introduction

The neural basis of cognitive vocal control underlying human speech and its evolutionary emergence in the primate lineage remain poorly understood. In humans, a key structure endowing volitional speech control is Broca's area in the inferior frontal lobe. Broca's area classically comprises areas 44 and 45 in the ventrolateral prefrontal cortex (vlPFC). These areas are instrumental for producing speech and language (*Friederici and Chomsky, 2017*; *Penfield and Roberts, 1959*); damage to these areas causes speech production aphasia. Recent evidence suggests that Broca's area may not directly regulate speech articulation but rather affect the cognitive preparation of speech (*Dronkers and Baldo, 2010*). For instance, cooling of Broca's area in awake neurosurgical patients slows speech without affecting articulation (*Long et al., 2016*).

Several lines of evidence suggest that the neural correlates of human speech could have evolved from a basic cognitive vocal control system already present in the nonhuman primate frontal lobe. In macaques, area 44 is located deep in the inferior arcuate sulcus (ASi), whereas area 45 is part of the ventral pre-arcuate region (VPA) of the vlPFC. Both areas have been identified as an anatomical homolog of Broca's area in the human brain (*Petrides, 2014*; *Petrides and Pandya, 1994*; *Petrides and Pandya, 2002*). Anatomically precise electrical stimulation in area 44 of the ASi of anesthetized monkeys elicits orofacial and laryngeal movements that are in the service of speech production in humans (*Petrides et al., 2005*). Moreover, in macaques trained to vocalize on command, neurons in vlPFC respond specifically in preparation of volitional calls (*Gavrilov et al., 2017*; *Hage and Nieder, 2013*; *Hage and Nieder, 2015*). Importantly, vlPFC neurons show a strong correlation between the onset of neuronal activity and the timing of vocal output, suggesting that the vlPFC is forming a decision signal for initiating vocalizations (*Gavrilov et al., 2017*).

In order to demonstrate 'volitional control', three criteria have to be fulfilled in unison (*Brecht et al., 2019*; *Nieder and Mooney, 2020*). First, responses need to be executed in consequence of an arbitrary instruction stimulus that is neutral in its value or emotional valence. This criterion is important to ensure that motor acts, and vocalizations in particular, are not elicited by internal (affective/motivational/arousal) status changes in the presence of food or predators (*Fischer and Price, 2017*; *Jürgens, 1979*). Second, responses need to be uttered in a manner that is temporally contingent to the instruction stimulus. Third, actions need to be reliably withheld in the absence of an instructive stimulus. In neuropsychological tests, this list of criteria is similarly applied to differentiate between volitional and affective/spontaneous responses in patients (*Cattaneo and Pavesi, 2014*; *Hopf et al., 1992*). For example, patients with facial paralysis due to damage of descending pathways from the motor cortex have considerable difficulty smiling or frowning on command, a condition called 'voluntary facial paresis'; nevertheless, they smile or frown spontaneously in response to their emotional states. Similar dissociations have been observed for vocalizations: some patients with neurological insults may lose volitional control of their speech, but can still laugh, scream, or groan when they are happy, frightened, or in pain. To determine the neuronal basis of different types of motor responses, we adopt a protocol that fulfils these criteria in the current study in order to distinguish volitional from spontaneous responses.

Despite advances in our understanding of vocal control mechanisms in primates, it is currently not known if vocalization-correlated neurons are part of a dedicated vocal network that specifically encodes the volitional preparation of vocalizations. After all, the primate PFC is regarded as the central executive of the brain (*Miller and Cohen, 2001*) and hosts a variety of cognitive functions necessary for motor planning and goal-directed action control (*Tanji et al., 2007*). One hypothesis therefore is that vlPFC neurons encode the preparation of various volitional actions, irrespective of the effector organs. In this case, such neurons would not only signal the initiation of vocalizations, but also of other actions, such as the preparation of hand movements. The alternative hypothesis predicts that neurons in the vlPFC show functional specialization for the volitional preparation of specific motor acts; according to this scenario, neurons would be functionally segregated and encode the initiation of only one type of motor act, for instance, of either vocalizations or hand movements. We tested these hypotheses and recorded single neuron activity from the vlPFC (the ventral pre-arcuate region, VPA, and in the fundus of the inferior arcuate sulcus, ASi) and the adjacent rostroventral premotor cortex (PMrv, part of area 6) of rhesus macaques trained to either vocalize or make a hand movement in response to arbitrary visual cues.

## Results

### Behavioral performance

We trained two rhesus monkeys in a computer controlled 'go/nogo' detection task. The monkeys either had to vocalize ('vocal' trials) or to respond manually by releasing a bar ('manual' trials) in response to the presentation of arbitrary visual stimuli to receive a reward (*Figure 1*). Two stimuli per trial types were used: a red cross or a blue square both cued vocalizations; a yellow ring or green square both prompted hand movements. Vocal trials and hand trials were presented in blocks within each session. The blocks switched after 25 correctly performed 'go' trials. Even block numbers comprised 'vocal trials', whereas odd block numbers consisted of 'manual' trials.

We collected data from 20 daily sessions from monkey A and 41 sessions for monkey P. On average, a session consisted of 4 blocks per condition (monkey A) or five blocks per condition (monkey P), respectively. In each session, monkey A uttered 84.2 ± 30.9 calls and monkey P elicited 104.5 ± 27.1 calls. First, we analyzed the monkeys' behavioral performance. To that aim, we used signal detection theory. To compare the performances for both response types ('vocal' and 'manual'), we calculated 'hit' rates, 'false alarm' rates, and d' values for each condition separately. The production of cued vocalizations was more difficult for the monkeys than the production of cued hand movements.

*Figure 2* shows a representative session by monkey P comprising of 78 'grunt' vocalizations and 84 'manual' responses. In the 'vocal' block, the monkey uttered 73 calls in response to the 'go' cue. However, the monkey also missed to vocalize in 108 'go' trials which resulted in a 'hit' rate of 39.3% (73 of 186 'go' trials). Not a single call was uttered during 'catch' trials (trials without a 'go'-cue) or

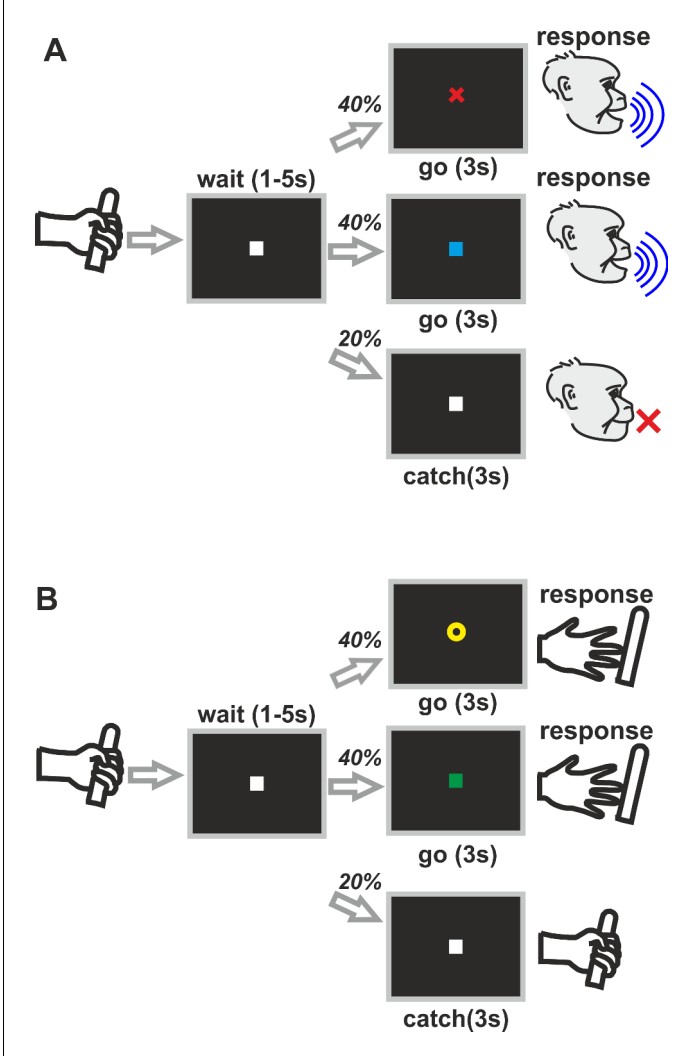

**Figure 1.** Experimental design. Two monkeys were trained in a 'go/nogo' detection task. In alternating trial blocks, the animals had either to vocalize in response to one of two arbitrary visual cues (blue square or red cross) (A) or to release a bar in response to two other visual stimuli (green square or yellow circle) (B).

during 'manual' trials. The remaining five vocalizations were uttered during the wait periods (three calls; 3.8%) and after cue offset of 'go' trials (two calls; 2.6%). During the 'hand' block, the monkey produced a high proportion of hand releases in response to the corresponding 'go' stimuli. In contrast to 'vocal' trials, the animal did not miss to respond in 'manual' trials, resulting in a 'hit' rate of 100%. In four trials, the monkey erroneously responded with a bar release in 'vocal' trials, resulting in a 'false alarm' rate of 2.2% (4/179). For this session, the obtained 'hit' and 'false alarm' values led to a mean d' sensitivity value of 3.1 and 5.4 in 'vocal' and 'manual' trials, respectively. This shows that the monkey produced each response type reliably and almost exclusively in response to the corresponding visual 'go' stimuli.

Throughout the sessions, both monkeys never missed to release the bar in response to the corresponding 'go' cue during 'manual' trials, which resulted in a 'hit' rate (HR) of 100% (*Figure 3A,B*). At the same time, the monkeys rarely erroneously released the bar in 'catch' and 'vocal' trials ('false alarm' (FA) of 12% and 5%, respectively). Because of the high 'hit' and low 'false-alarm' rates, the d'-sensitivity values for cued hand movements were well above the threshold of 1.5 during 'manual' trials (4.3 ± 0.7 monkey A; 5.3 ± 0.7 in monkey P) (*Figure 3C,D*).

In 'vocal' trials, both monkeys missed to vocalize in response to the 'go' cue in about 60% of the trials (averaged hit rate: 38% monkey A; 55% monkey P). However, both animals almost never called

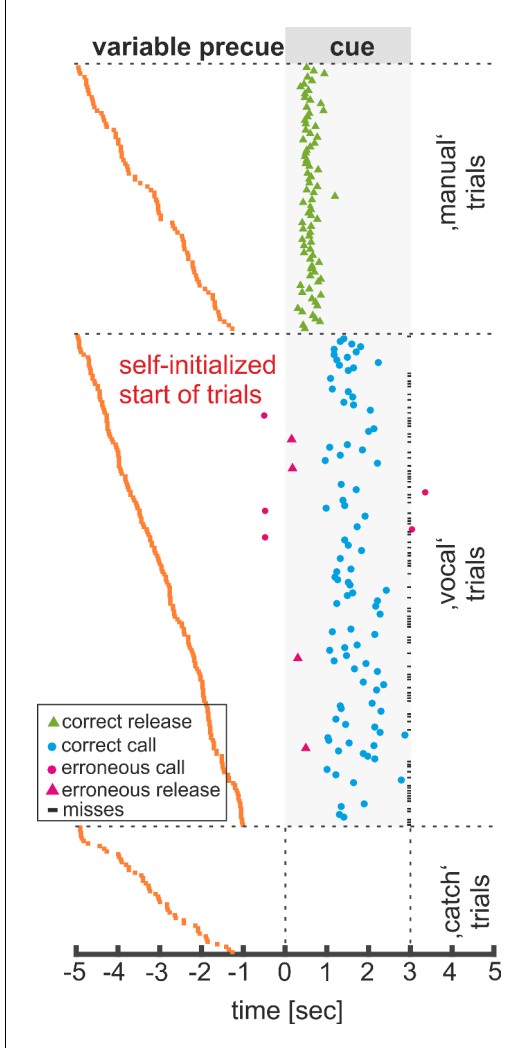

**Figure 2.** Example of a single session of monkey P. Responses in 'vocal', 'manual' and 'catch' trials are sorted according to the length of the 'pre-cue' signal. Each line represents a single trial; blue circles indicate vocal onsets and green triangles indicate timing of the bar release. Pink circles and triangles indicate wrong responses. 'go' trials ignored by the monkey ('misses') are marked with a horizontal black bar at trial end.

in 'catch' trials or the 'manual' trials. Decent 'hit' rates and very low 'false alarm' rates in the 'vocal' trials cause above threshold discrimination sensitivity (d') performance on all recording sessions (average d'-values of 2.4 ± 0.5 in monkey A, and 3.0 ± 0.6 in monkey P).

As a second behavioral parameter, we also analyzed reaction time (*Figure 3E,F*). While the monkeys responded on average within the first 500 ms after the 'go' cue onset in the 'manual' task (on average 0.51 s and 0.54 s for monkey A and monkey P, respectively), they uttered cued vocalizations only over a second later (on average 2.00 s for monkey A and 1.64 s for monkey P). These relatively long call reaction times are comparable to our previous reports with two different monkeys (*Gavrilov et al., 2017*; *Hage and Nieder, 2013*).

## Response type correlated neurons

While the animals performed the task, we recorded 863 single neurons (286 neurons from monkey A and 577 neurons from monkey P) in two regions of the ventrolateral PFC (the ventral pre-arcuate region, VPA, and inside the inferior arcuate sulcus, ASi) and the adjacent rostroventral premotor cortex (PMrv, part of area 6rv) (*Figure 4*). The main aim was to investigate whether neurons in these frontal areas would differentiate between cued initiation of vocalizations and hand movements.

We first cleared the data from putative confounds with eye-movement-related activity. Since we measured the eye movements of the monkeys that were not required to maintain fixation, neurons showing eye-movement-related activity could be identified (see Materials and methods for details). We found that 37.3% of all recorded neurons (322/863) exhibited significant eye movement-related activity, and a further 4.2% of the neurons (36/863) showed significant fixation-related activity. All these neurons were excluded from further analyses.

We then identified neurons that showed activity which was correlated with the instructed initiation of vocalizations and/or hand movements. To that aim, we used the neurons' firing rates after 'go' cue presentation in a 450 ms interval immediately before the monkeys' responses (vocalization or hand movement). This time window was adjusted to the monkeys' short average reaction times during 'manual' trials (individual trials with response latencies shorter than 450 ms were discarded) and thus allowed a comparison of premotor activity for both 'manual' and 'vocal' trials. Neurons that significantly modulated their firing rate in this premotor time window prior to hand movements during 'manual' trials compared to baseline activity were defined as hand-correlated neurons (Wilcoxon signed rank test; p<0.05). Similarly, neurons that changed their activity during the same time interval prior to vocalizations in 'vocal' trials were defined as vocalization-correlated neurons. Figure 5 shows the time course of the neuronal activity of six example neurons from the three recording areas together with the averaged and normalized activity of the cell population corresponding to the

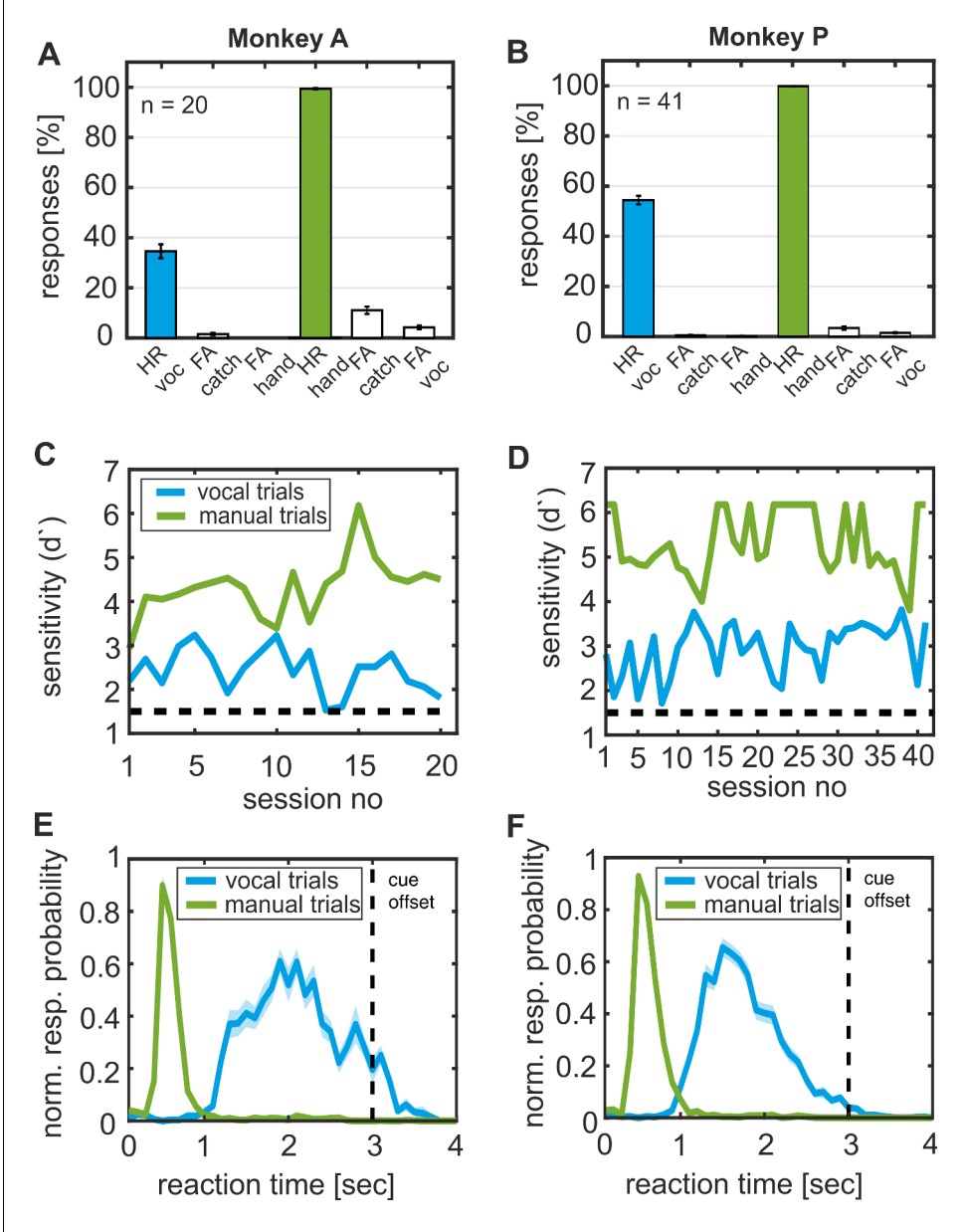

**Figure 3.** Behavioral performance. (**A**, **B**) Distribution of 'hit' and 'false alarm' rates for each response type and animal separately, averaged over 20 sessions for Monkey A and 41 for monkey P. (**C**, **D**) Sensitivity of signal detection for 'vocal' and 'manual' trials indicated by the d' prime value. The dotted line indicates the detection threshold of 1.5. (**E**, **F**) Response probability of 'vocal' and 'manual' responses in the corresponding trial type. Normalized and averaged response over sessions are shown. Shaded areas indicate first and third quartiles.

respective example neurons. In this figure, activity in both 'vocal' and 'manual' trials (only correct trials) is aligned relative to response onset (black vertical line in the histograms). *Figure 5A,C,E* depicts three vocalization-correlated neurons, one for each recording area (i.e. PMrv, ASi, and VPA), that show a significant increase in discharge rate prior to vocal onset. Interestingly, the firing rates of the same neurons hardly changed in 'manual' trials when the monkeys prepared a bar release (note that some neurons were selective to both vocalization and hand movements, as later discussed in *Figure 6*). Overall, we found 20% (26/130), 24% (55/228) and 24% (121/505) vocalization-correlated neurons in PMrv, ASi, and VPA, respectively. Approximately one-third of vocalization-correlated neurons recorded in PMrv showed increased activity (9/26: 34.6%), whereas the remaining two-thirds of

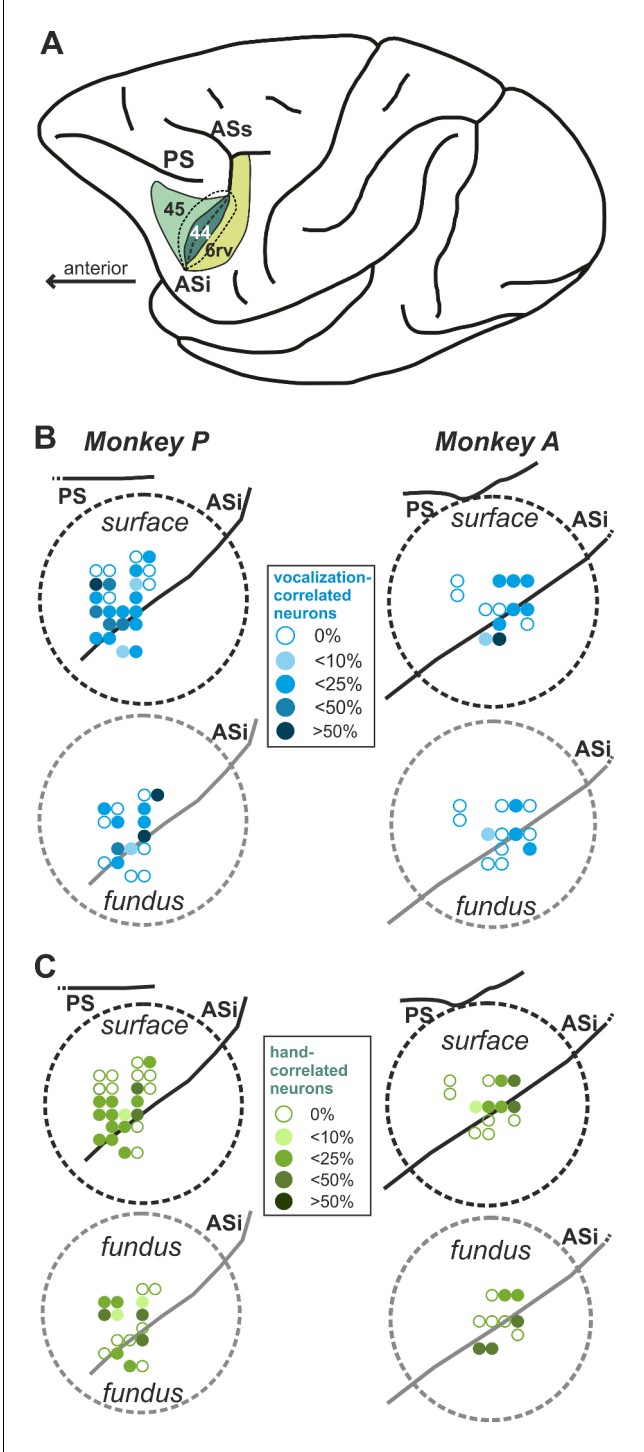

**Figure 4.** Recording sites in the inferior frontal lobe of both monkeys. (**A**) Lateral view of the left hemisphere indicating the recording area: parts of the rostroventral premotor cortex (PMrv of area 6), the fundus of the inferior arcuate sulcus (ASi) encompassing area 44, and the ventral pre-arcuate region containing parts of area 45. The inferior arcuate sulcus is unfolded, with dotted lines marking the transition from the cortical surface to the gyral walls. PS, principal sulcus; ASi, inferior arcuate sulcus; ASs, superior arcuate sulcus (**B, C**) Precise recording sites inside each recording chamber (dotted circles). Recording sites with a depth >6 mm were defined as ASi sites. For better overview, recording sites within the fundus of the inferior AS (area 44) are depicted offset on the right side of each chamber. The proportion of vocalization-correlated neurons (**B**) in relation to all neurons recorded at a

*Figure 4 continued on next page*

*Figure 4 continued*

specific recording site is coded with different shades of blue. Similarly, the proportion of hand-correlated neurons
(C) in relation to all neurons recorded at a specific recording site is coded with different shades of green color.

vocalization-correlated neurons exhibited decreased activity (17/26: 65.4%) prior to vocal onset. In the two other areas, the proportion of excited and suppressed neurons was roughly the same. In ASi, we found 56% excited neurons (31/55) and 44% suppressive neurons (24/55). In VPA, we detected 45% excited neurons (55/121) and 55% suppressed neurons (66/121).

A comparison of neuronal latencies of vocalization-correlated neurons revealed no significant differences between the three areas (median latency for PMrv: 1068 ms before vocal onset; median latency for ASi: 1099 ms; median latency for VPA: 871 ms; Kruskal-Wallis test, p=0.12).

To summarize activity profiles from excited and suppressed neurons, the normalized activity of all suppressed neurons was rectified relative to the baseline activity so that negative deflections were transferred into positive deflections of equal magnitude. The average normalized discharge rates of all vocalization-correlated neurons in the respective recording areas are shown in *Figure 5B,D,F*. They show continuously rising activity toward call onset while hardly any activity change can be seen for the preparation of hand movements.

We also detected neurons that signaled the preparation of hand movements. Three such neurons are depicted in *Figure 5G,I,K*. All three neurons show an increase in firing rates prior to hand responses while premotor activity remained unchanged in 'vocal' trials. In total, we found 10% (13/130), 13.6% (31/228), and 16.8% (85/505) hand-correlated neurons in PMrv, ASi, and VPA, respectively. The proportion of neurons that increased or decreased its activity prior to the bar release varied widely between the three areas, ranging from 51% of excited neurons in ASi, followed by 59% in VPA and 85% in PMrv. The averaged and normalized activities of all hand-correlated neurons (excited and suppressive) for the corresponding recording areas are shown in *Figure 5H,J,L* and show a relatively rapid increase in activity shortly before the cued hand movement, but not in preparation of calls.

Similar to vocalization-correlated neurons, a comparison of neuronal latencies of hand-correlated neurons revealed no significant difference between the areas (median latency for PMrv: 252 ms prior to response onset; median latency for ASi: 316 ms prior to response onset; median latency for VPA: 257 ms prior to response onset; Kruskal-Wallis test p>0.66).

The proportion of vocalization-correlated neurons in each area was significantly larger compared to the proportion of hand-correlated neurons (vocalization-correlated neurons overall (202/863) and hand-correlated neurons overall (129/863), p<0.001; separated by area: PMrv vocalization-correlated neurons (26/130) and hand-correlated neurons (13/130), p<0.05; ASi vocalization-correlated neurons (55/228) and hand-correlated neurons (31/228), p<0.01; VPA vocalization-correlated neurons (121/505) and hand-correlated neurons (85/505), p<0.01; all $\chi^2$ tests).

To investigate whether cued vocalizations and hand movements were decoded by the same population of neurons or rather by two separated populations, we compared the frequencies of only hand-correlated and only vocalization-correlated neurons with the frequency of neurons that encoded both response preparations, for each area separately. More specifically, we explored if we would find more neurons that encoded the preparation of both response types than expected by chance based on the frequencies of neurons that encoded only the preparation of hand movements or vocalizations, and additionally show activity profiles moving in the same direction (i.e. excitation and suppression in vocalization as well as in 'manual' trials).

In PMrv, the initiation of the 'vocal' response was represented by 20% of all neurons recorded in this area, and the initiation of the 'manual' response by 10% of all neurons. Chance predicts that 2% of all neurons (0.20 × 0.10) would encode the initiation of both response types, which correlated with the 1.5% (2/130; *Figure 6A*) of all neurons encoding both response types (p=0.27, binomial test). In ASi, 3.3% of all neurons (0.24 × 0.136) that encoded the initiation of both response types could be expected by chance. Empirically, we found 6.1% (14/228; *Figure 6B*), which is slightly more than expected by chance (p=0.018, binomial test). In VPA, chance frequency of neurons encoding the initiation of both cued vocalizations and cued hand movements was 4% of all neurons (0.24 × 0.168). With a measured frequency of 6.5% (33/505; *Figure 6C*), slightly more VPA-neurons than expected by chance were found (p<0.001, binomial test). Overall, however, the overlap between the

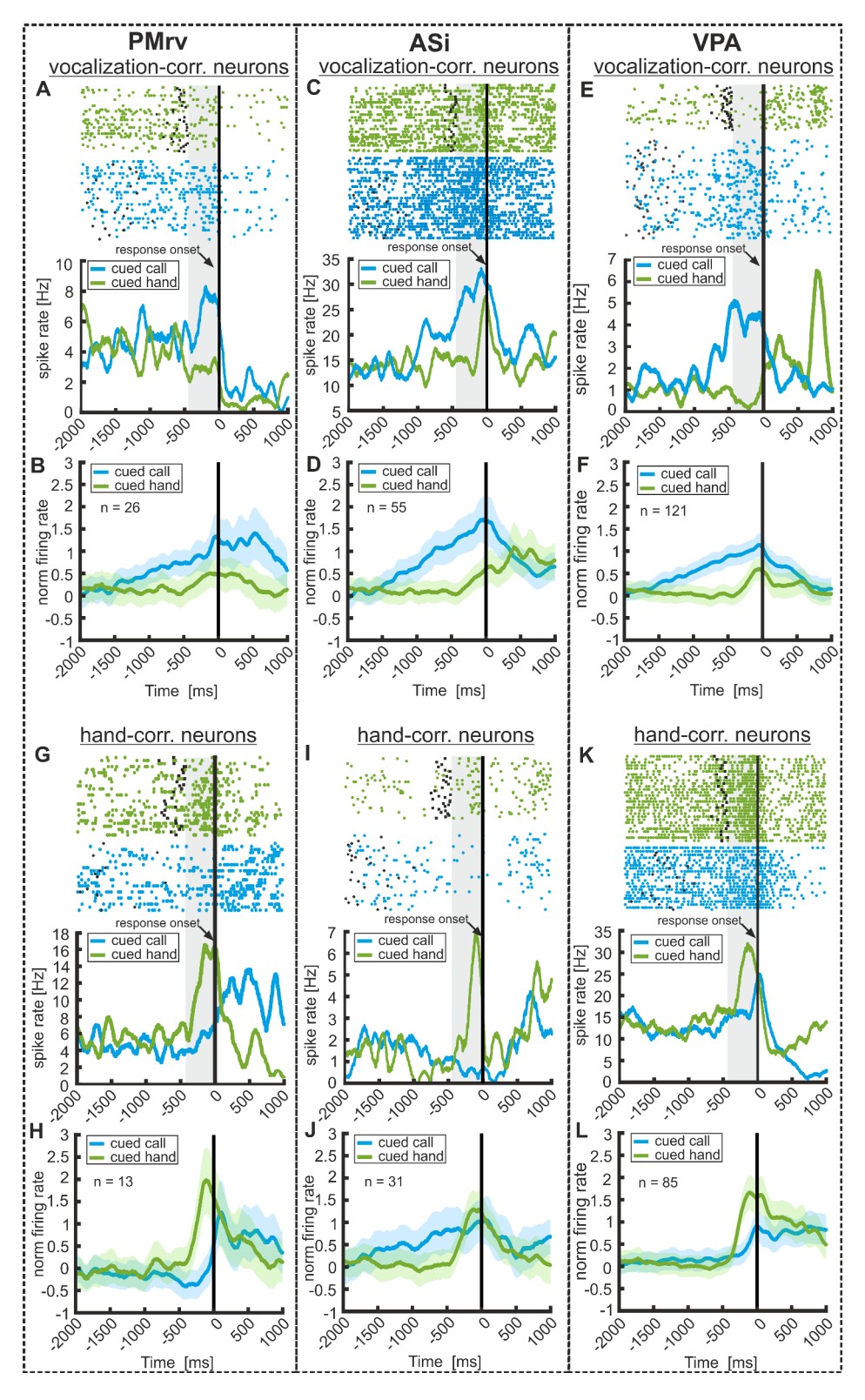

**Figure 5.** Neuronal responses to the preparation of volitional vocalizations and hand movements in PMrv (left), ASi (middle), and VPA (right). Responses of three example vocalization-correlated neurons and three hand-correlated neurons in 'vocal' and 'manual' trials as well as averaged and normalized population responses for the corresponding examples. Responses are recorded in PMrv (left column), ASi (middle column), and VPA (right column). (**A, C,E**) Three example vocalization-correlated neurons that show a significant increase of neuronal activity during 450 ms before the instructed vocalization

*Figure 5 continued on next page*

*Figure 5 continued*

compared to baseline (450 ms before 'go' cue onset). At the same time, the neurons show only minor changes in activity preceding the response in 'manual' trials. Upper panel show the raster plot, black dots and asterisks indicate the 'go' cue onset in the 'vocal' and 'manual' trials, respectively. The lower panel represent the corresponding spike density histogram averaged and smoothed with a Gaussian kernel (150 ms) for illustration. The vertical black line indicates the response onset. Vocal responses have an average duration of 97 ± 40 ms. (B,D,F) Averaged and normalized activity of all vocalization correlated neurons recorded in the corresponding area. These-neurons show a significantly different activity prior to cued 'vocal' responses compared to cued 'manual' responses (activity tested in 450 ms before response onset; Mann Whitney U test: p<0.001). (G,I,K) Three example hand-correlated neurons showing a significant increase in firing rate before the instructed 'manual' response compared to baseline activity. The activity in 'vocal' trials remains at baseline level. (H,J,L) Averaged and normalized activity of all hand-movement-correlated neurons recorded in the corresponding area. Hand-correlated neurons show a significant change in firing rate only preceding 'manual' responses but not preceding 'vocal' responses (activity tested in 450 ms before response onset; Mann Whitney U test: (H,J) p<0.001, (L): p=0.115). Shaded area around the curves depicts the standard error of the mean (s.e.m.).

populations of vocalization- and hand-correlated neurons was negligible, suggesting that largely separate neuron populations encoded either cued vocalizations or hand movements.

## Cued versus spontaneous responses

Both animals now and then called and released the bar spontaneously in between trials or in the absence of a 'go' cue. When contrasting cued responses with spontaneous responses, we found a clear difference in firing rates prior to response onset. Two example vocalization-correlated neurons are depicted in *Figure 7D,E* (right). Both neurons show a significant change in activity only prior to cued but not prior to spontaneous vocalizations. Such a significant difference between cued and spontaneous vocalizations was also found for the 14 vocalization-correlated neurons for which enough spontaneous vocalizations could be recorded (Wilcoxon signed rank test, p<0.01). *Figure 7F* shows the averaged normalized firing rates of these neurons (normalized suppressed activity was again rectified relative to the baseline). This finding argues that the coding of cued vocalizations is different from the coding of spontaneous vocalizations; only cued vocalizations elicited activity deviating from baseline (*Figure 7F*).

Interestingly, also hand-correlated neurons showed different activity profiles prior to cued and spontaneous bar releases. While the discharge rate increased/decreased significantly shortly before the instructed hand movements, the activity stayed at baseline level shortly before spontaneous bar releases (*Figure 7A,B*). The averaged normalized firing rates of 44 hand-correlated neurons are shown in *Figure 7C*, and they differed significantly between the cued and spontaneous conditions (Wilcoxon signed rank test; p<0.001). These results indicate that the activity of these neurons might encode the 'willingness' to initiate a specific response. However, since the two response types (calls and hand movements) are produced by completely different neuronal networks which require a different preparatory period, the two response types might be encoded by two separate populations of neurons.

## Quality and temporal evolution of response type coding

To quantify how well calls and hand movements could be discriminated based on the distribution of each vocalization- or hand-correlated neuron's firing rates in 'vocal' and 'manual' trials over time, we performed a sliding ROC analysis. The area under the ROC curve (AUROC) is a measure of the separation of two distributions, with 0.5 indicating complete overlap, and both 0 and 1 indicating perfect separation. Hand-correlated neurons recorded in PMrv show a stronger coding of the 'manual' response compared to vocalization-correlated neurons in PMrv that encode the' vocal' response (*Figure 8A*; averaged difference from 0.5; manual: 0.23 ± 0.15; vocal: 0.13 ± 0.08; p=0.042; Mann–Whitney U-test). The opposite was true for the neurons recorded in ASi (*Figure 8B*). Here, the 'vocal' response coding by vocalization-correlated neurons was stronger than the 'manual' response coding by hand-correlated neurons (averaged difference from 0.5; manual: 0.12 ± 0.08; vocal: 0.17 ± 0.10; p=0.046; Mann–Whitney U-test). The strength of response type coding was similar for both hand- and vocalization-correlated neurons recorded in VPA (*Figure 8C*; averaged difference from 0.5; manual: 0.19 ± 0.10; vocal: 0.17 ± 0.09; p=0.106; Mann–Whitney U-test).

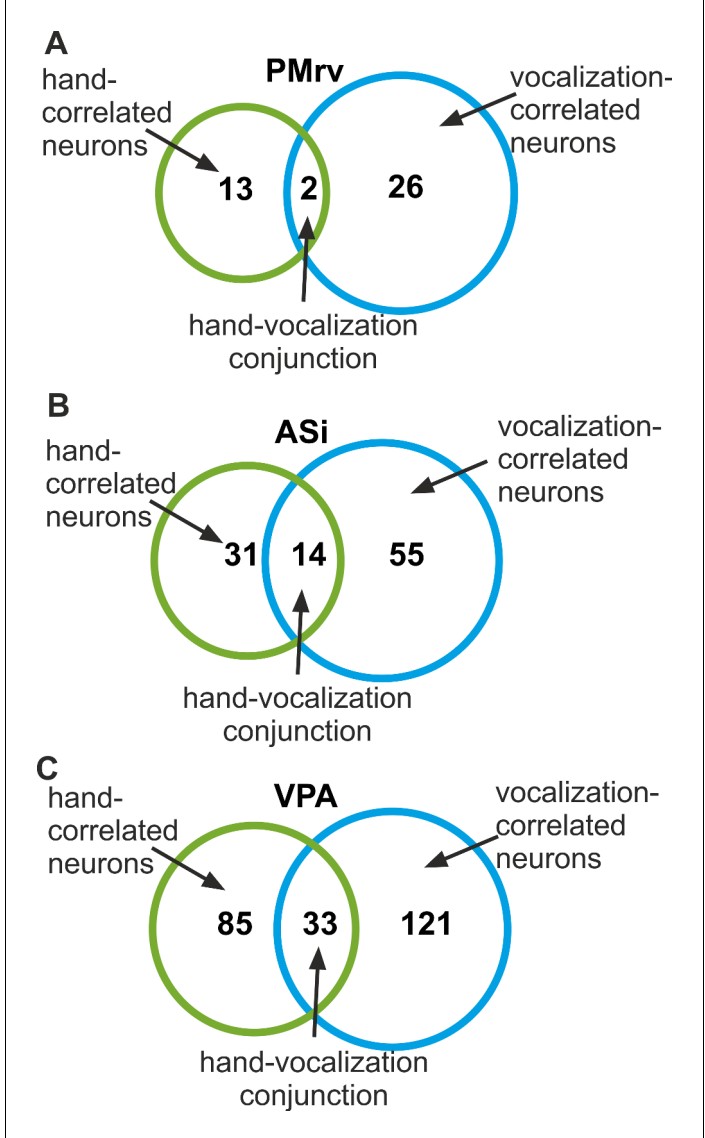

**Figure 6.** Proportions of hand- and vocalization-correlated neurons recorded in PMrv (**A**), ASi (**B**), and VPA (**C**). The Venn diagrams depict the number of hand-correlated neurons and vocalization-correlated neurons that encode only one type of the response ('vocal' or 'manual') or both response types at the same time.

## Neuronal population coding

Furthermore, we explored whether and how the entire population of recorded neurons, irrespective of any response preferences, encoded the two different response types ('manual' and/ or 'vocal'). Therefore, we performed a multi-dimensional state space analysis (Gaussian-Process Factor Analysis, GPFA) (*Yu et al., 2009*) on a population of pseudo-simultaneously recorded neurons for each brain area separately (PMrv: n = 36; ASi: n = 81; VPA: n = 180; for Details see Materials and methods). This approach extracts trajectories from the spiking activity of a neuronal population in individual trials. Such trajectories reflect the instantaneous firing rates of the respective neuronal population as they evolve over time.

*Figure 9A,B,C* depict each two average population trajectories for 'vocal' and 'manual' trials in a space defined by the top three most meaningful dimensions in the respective brain areas. To evaluate the temporal evolution of population activity in each brain area and to different trial types, we measured Euclidian distances between trial trajectories corresponding to the same trial type (within-response coding) and different trial types (cross-response coding) (*Figure 9D,E,F*). For two of the

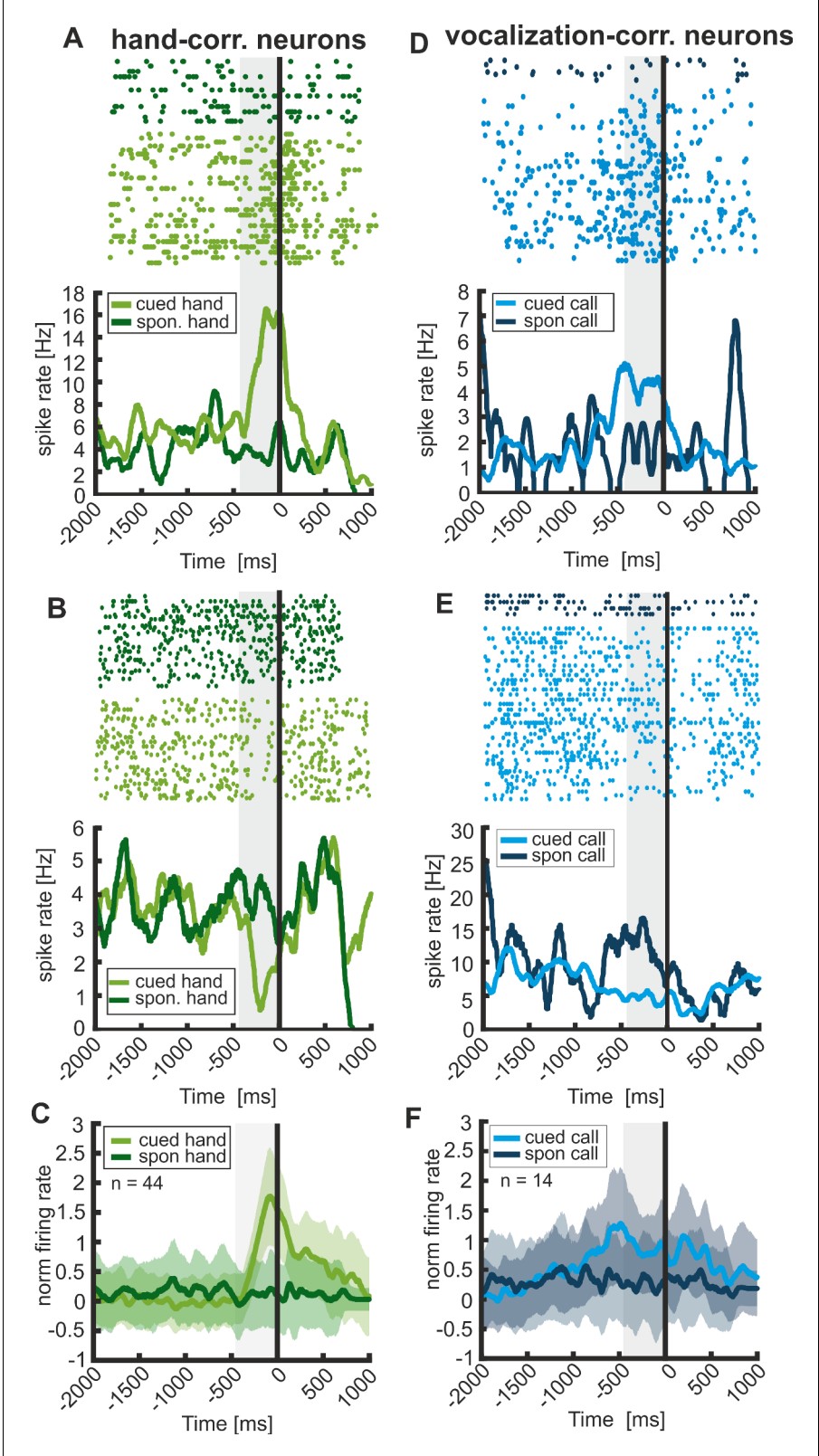

**Figure 7.** Response type correlated activity of vocalization- and hand-correlated neurons preceding cued and spontaneous responses. Left and right column represent the activity in 'cued' and 'spontaneous' trials of two example hand- and vocalization-correlated neurons, respectively (**A,B,D,E**). Vocalization-correlated neurons show significantly increased or decreased activity prior to instructed calls but not prior to spontaneously uttered

*Figure 7 continued on next page*

*Figure 7 continued*

vocalizations. (**C,F**) Averaged and normalized population activity of 44 hand-correlated and 14 vocalization-correlated neurons that were recorded during 'cued' and 'spontaneous' trials. Hand-correlated neurons show a significantly different activity prior to cued 'manual' responses compared to 'spontaneous manual' responses (activity tested in 450 ms before response onset; Wilcoxon signed rank test: p<0.001). Similarly, vocalization-correlated neurons show a significantly different activity prior to 'cued vocal' responses compared to 'spontaneous vocal' responses (Wilcoxon signed rank test; p<0.01). Shaded area around the curves depicts s.e.m.

three areas (namely ASi and VPA), a large inter-trajectory distance was apparent only between 'vocal' and 'manual' trials (cross-response coding), indicating that both response types are represented by different activity profiles. For both ASi and VPA, the increase in inter-trajectory distance occurred around a second before response onset, which was mainly caused by the increased activity of vocalization-correlated neurons. At a time point of about 450 ms before response onset, there was a decrease in inter-trajectory distance, possibly caused by the increased activity of hand-correlated neurons. The time course as well as the strength of activity in the whole populations of neurons recorded in ASi and VPA was consistent with the findings based on response type correlated neurons (vocalization- and hand-correlated neurons) only. While we also found vocalization- and hand-correlated neurons in PMrv, the whole population of neurons recorded in this area did not seem to differentiate between the two response types. In sum, these results indicate a stronger involvement of ASi and VPA in the representation and preparation of different cued motor responses compared to PMrv.

## Discussion

The main aim of this study was to investigate if and how neurons in the ventrolateral frontal lobe (ventral pre-arcuate region, VPA, and inferior arcuate sulcus, ASi, of the vlPFC, and rostroventral premotor cortex, PMrv), core areas of Broca's language production region in humans, would differentiate between cued initiation of vocalizations and hand movements indicating volitional control. In monkeys trained to elicit vocal and manual responses on command, we found the preparation of both action types (vocal and manual actions) represented by selective neurons in the three anatomical areas. In correlation with the monkeys' much faster reaction times for hand movements, hand-correlated neurons showed drastically shorter preparatory activity. Interestingly, the responses of selective neurons were specific to the volitional initiation of either action, indicating that these neurons' activity did not signify general motor preparation but only volitional motor plans. Moreover, neurons were selective to the volitional preparation of either vocalizations or hand actions, rarely both. This suggests distinct, slightly overlapping networks of neurons selective to vocal and manual action preparations. A higher proportion of vocalization-correlated neurons in all three brain areas suggests a specialization of the vlPFC for initiating volitional vocalizations.

### Volitional vocalizations and hand movements

The alternating task protocol allowed us to compare the behavioral characteristics of volitional vocalizations and hand movements under equal conditions. Although both monkeys performed the vocalization and manual trials reliably, all measured behavioral parameters argue that volitional vocalizations were considerably more demanding for the monkeys than instructed hand movements. The monkeys showed a perfect 'hit' rate of 100% for 'manual' trials, whereas average 'hit' rate in 'vocal' trials was less than 50%. On the background of similarly low 'false alarm' rates in 'catch' trials, this resulted in a discriminability sensitivity (d') that was almost twice as high and therefore much better for volitional hand movements compared to cued vocalizations. In addition, both monkeys elicited hand movements much more quickly than vocalizations. While hand movements were performed after about half a second, vocalizations were only uttered after reaction times that were almost four times as long. Thus, both in terms of performance and timing, instructed vocalizations were drastically more difficult for the monkeys. Similar findings for macaques were reported in other behavioral studies that compared instructed hand movements and vocal utterances and found a behavioral advantage for hand movements (*Sutton et al., 1981*; *Hage et al., 2016*; *Koda et al.,*

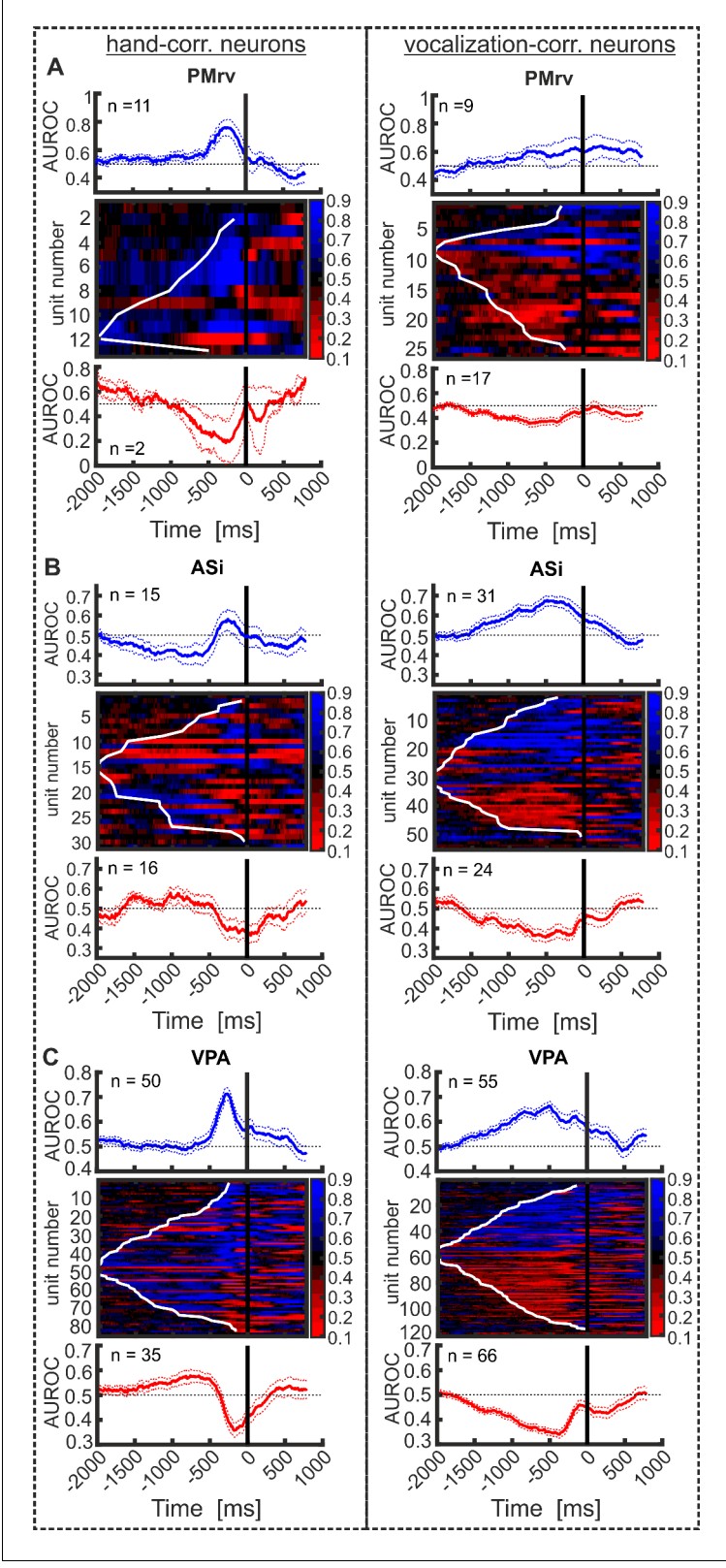

**Figure 8.** Quality and temporal evolution of response correlated activity for all hand-correlated neurons (left column) and all vocalization-correlated neurons (right column), recorded in PMrv (**A**), ASi (**B**), and VPA (**C**). Neurons are sorted based in their activity profile (increased or decreased FR) in the pre-response period compared to baseline and the latency of response type coding. Top: average AUROC values of all hand- (left) or vocalization-

*Figure 8 continued on next page*

*Figure 8 continued*
correlated neurons (right) increasing their FR prior to 'manual' response. Middle: AUROC values of each individual hand- or vocalization-correlated neuron from 2000 ms before to 1000 ms after response onset. The black line at 0 ms indicates the response onset. The white line marks each neuron's latency of response type discrimination. Bottom: average AUROC values of all hand- (left) or vocalization-correlated neurons (right) showing decreased activity prior to response onset. Dashed lines represent the s.e.m.

*2018*). Thus, the capability to produce volitional vocalizations seems to be evolutionarily delayed in primates relative to volitional hand movements.

## The vlPFC involved in manual and vocal action planning

The primate vlPFC is known to be involved in the planning of different motor acts. On the one hand, the vlPFC is involved during sensorimotor transformation required for hand actions and gestures (*Simone et al., 2015*; *Yamagata et al., 2012*). Neurons in the vlPFC not only encode object properties (*Rainer and Miller, 2000*; *Ramirez-Cardenas et al., 2016*) and mnemonic information (*Funahashi et al., 1989*; *Rainer et al., 1999*; *Eiselt and Nieder, 2013*), but also contribute to the planning and control of hand actions across various contexts, such as during light vs. darkness, and memory vs. visually-guided actions (*Simone et al., 2015*). This suggests that the vlPFC integrates contextual information and generates the goal of the intended action during action planning and execution. This information may subsequently be distributed to the ventral premotor cortex (PMv)

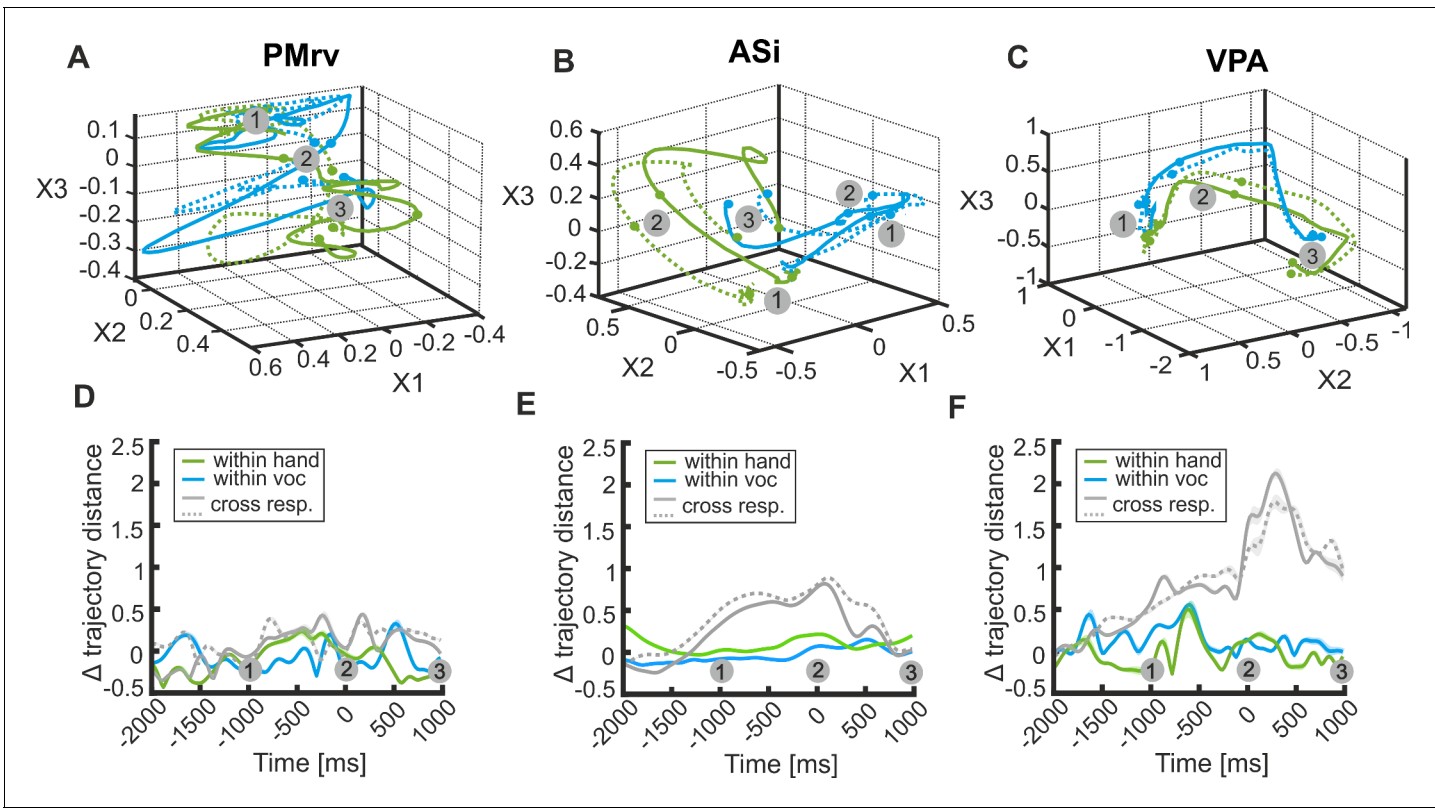

**Figure 9.** Temporal evolution of pre-response activity in the whole populations of neurons recorded in PMrv (left), ASi (middle), and VPA (right). (A–C) Gaussian Process Factor Analysis delineating the state-space of neuronal population activity (PMrv: n = 36 neurons; ASi: n = 81 neurons; VPA: n = 180 neurons) over time for 'manual' (green) and 'vocal' trials (blue), respectively. Solid and dashed lines of the same color represent each one half of the trials from the same condition. Each trajectory represents the average of 100 repetitions of GPFA performed on randomly selected 17 trials (VPA) or 15 trials ASi and PMrv for each condition and each neuron. Dots and numbers indicate time point 1000 ms before response onset (1), the response onset (2) and time point 1000 ms after response onset (3). (D–F) Averaged inter-trajectory Euclidean distance over time, as a measure for within-response coding (colored lines) and cross-response coding (gray lines). Shaded area around the curves represents the s.e.m.

and inferior parietal lobule (IPL), brain areas involved in visuo-motor transformation of hand actions (*Bonini et al., 2010*; *Fogassi et al., 2005*) to which the vlPFC is connected (*Borra et al., 2011*; *Gerbella et al., 2010*), in order to successfully perform goal-directed hand actions via output to the primary motor cortex (*Battaglia-Mayer and Caminiti, 2019*; *Borra et al., 2017*; *Hoshi and Ishida, 2015*). The increased dexterity of forelimb, hand, or finger voluntary actions necessary for gestures also correlates with the expansion of direct cortico-motoneuronal connections along primate evolution (*Lemon, 2008*).

Besides hand movements, the vlPFC in nonhuman primates is also implicated in orofacial movements and the planning of vocal output (*Eliades and Miller, 2017*; *Nieder and Mooney, 2020*). Electrical stimulation of area 44 in the ASi, the suggested homologue of the cortex covering the pars opercularis in Broca's region in the human brain (*Frey et al., 2014*), elicits orofacial movements in anaesthetized macaques (*Petrides et al., 2005*). The strongly interconnected areas 44 and 45 within the vlPFC contain neurons that selectively discharge when trained macaques prepare an instructed vocalization (*Gavrilov et al., 2017*; *Hage and Nieder, 2013*; *Hage and Nieder, 2015*). Moreover, area 44 is strongly connected with ventral premotor cortex (ventral area 6) in which neurons exhibit responses related to mouth movements and vocalizations (*Coudé et al., 2011*; *Fornia et al., 2018*; *Shepherd and Freiwald, 2018*), but also the rostral part of the inferior parietal lobule in which neurons also exhibit responses related to mouth movements (*Leinonen and Nyman, 1979*). The primate lateral frontal cortex is connected with the posterior temporal region and the adjacent inferior parietal lobule via three major white matter pathways that subserve auditory-vocal processing: the arcuate fasciculus connects the posterior temporal region (referred to as Wernicke's area in humans and involved in the comprehension of language) with the inferior frontal gyrus (IFG) (incl. Broca's area in the language dominant hemisphere of humans). In addition, the third branch of the superior longitudinal fasciculus (SLF III) links rostral inferior parietal lobule with areas 6 and 44, whereas the second branch of the superior longitudinal fasciculus (SLF II) links area 45 with posterior inferior parietal lobule (i.e. angular gyrus) (*Barbeau et al., 2020*; *Petrides, 2014*; *Petrides and Pandya, 1984*). This anatomical segregation suggests distinct functional roles of these major pathways in audio-vocal processing.

As a classical multi-modal association and executive brain area receiving not only visual but also auditory information, the vlPFC is a well-situated hub for audio-vocal transformation (*Balezeau et al., 2020*; *Hage and Nieder, 2015*; *Hwang and Romanski, 2015*), top-down cognitive control (*Bichot et al., 2015*; *Vallentin et al., 2012*; *Viswanathan and Nieder, 2015*), and motor learning (*Petrides, 1982*; *Petrides, 2005*). Further evolutionary innovations for vocal control include elaborations of the laryngeal motor cortex (LMC; *Simonyan, 2014*) and projections from the LMC directly to phonatory and respiratory motoneuron pools as the dominant brain system for human speech and language (*Hage and Nieder, 2016*; *Nieder and Mooney, 2020*).

## Segregated vlPFC networks for vocal and manual control

Based on single-neuron recordings in parallel to behavior, we can trace these differences between these two types of volitional responses back to premotor activity in the inferior frontal lobe. The targeted prefrontal areas ASi and VPA, as well as the premotor area PMrv, do not seem to encode the general preparation of volitional acts irrespective of the effector organs. Rather, we found clear specificity of neurons in relation to vocal and manual preparatory activity, particularly in ASi and VPA of the vlPFC. This indicates segregated networks within the vlPFC that are part of a specialized vocal and manual control network.

Because the production of volitional vocalizations was cognitively much more demanding for the monkeys (as indicated by the reaction times and hit rates; see also *Hage et al., 2016*), the monkeys were kept motivated in the' vocal' trials by providing more reward than in the 'manual' trials (in which smaller and larger rewards were randomly assigned to counterbalance expectation). However, reward expectancy is unlikely to have caused the observed coding difference: at the behavioral level, the monkeys responded much faster in 'manual' trials in which the overall amount of reward was smaller (the opposite effect would have been expected if the amount of reward determined the reaction times). At the neuronal level, premotor activity in 'vocal' and 'manual' trials reached comparable peak activity prior to response onset. While we cannot exclude that reward expectation might have had a modulatory influence, differences in reward expectation cannot explain the observed coding differences.

Our finding of manual responses in the vlPFC concurs with a recent functional magnetic resonance imaging (fMRI) study in macaques. In monkeys that were trained to perform either hand grasping or ingestive mouth movements in the scanner, comparable moderate blood oxygen level dependent (BOLD) responses during the respective hand and mouth movements in area 44 were measured (*Sharma et al., 2019*). Our single neuron recordings suggest a more vivid involvement of these regions during volitional and potentially communicative face movements, such as vocalizations (*Hage and Nieder, 2013*) and lip smacking (*Petrides et al., 2005*; *Shepherd and Freiwald, 2018*). The specialized network for vocal call initiation might have been exploited and adapted over the course of primate evolution to serve as essential constituent of the human-specific speech and language system.

A specialization of the vlPFC in selecting different types of motor acts was recently also reported in the human brain based on fMRI. In the study by *Loh et al., 2020*, subjects were required to select between competing motor acts: manual, orofacial, nonspeech vocal, and speech vocal actions. It was found that area 44 reacted effector-dependent and was specifically involved in the selection between competing orofacial acts, as well as nonspeech vocal and speech vocal acts, but not during conditional manual responses. In contrast, area 45 was specifically recruited during the selection of both nonspeech and speech vocal responses, but only during learning (*Loh et al., 2020*). Our study in macaques confirms the role of area 44/45 in the cognitive control of vocal acts. However, in contrast to the study by *Loh et al., 2020*, we also found strong neuronal responses prior to hand actions in the vlPFC, albeit based on a smaller proportion of hand-correlated neurons. This discrepancy may be due to methodological differences between the two studies. In particular, the spatio-temporal resolution of the BOLD signal is limited compared to single-cell recordings. Also, the BOLD signal is less sensitive to detect a relatively small proportion of hand-correlated neuronal activity. Alternatively, the human area 44/45 may exhibit more pronounced neuronal specializations supporting the speech and language system, and a corresponding reduction of manual activity that is still lacking in old-world monkeys.

## Anatomically intermingled but functionally segregated action preparation networks in relation to language evolution

Our results in the monkey vlPFC also speak to the proposed evolutionary origins of the human language system. The two dominant hypotheses assume that the foundations for language evolution lay either in the gestural (visual-manual) or the vocal (audio-vocal) domain (*Kendon, 2017*; *Fitch, 2010*). The key question is whether one or the other of these hypotheses shows more cognitive pre-adaptations required for the emergence of language. One of these necessary pre-adaptations of a flexible communicative system is volitional control over motor acts (*Ackermann et al., 2014*; *Balter, 2010*; *Ghazanfar et al., 2019*) as defined in the introduction. The monkeys' hand movements easily suffice these criteria for volition. In addition, the current and other recent behavioral studies showed that monkeys can also produce vocalizations volitionally, albeit only with considerable cognitive effort (*Hage et al., 2013*; *Hage et al., 2016*; *Pomberger et al., 2019*). At a neuronal level, the current comparison of cued versus spontaneous vocalizations and hand movements showed impressive and highly significant activity differences; neuronal activity in the frontal lobe remains at baseline level for the monkeys' spontaneous and non-goal directed vocalizations and hand movements but is strongly modulated for volitional and goal-directed responses. This categorical distinction strongly argues that the neuronal coding of cued vocalizations is different from the coding of spontaneous vocalizations.

Within the monkey vlPFC, we found a co-occurrence of neurons encoding the preparation of different types of volitional motor acts. While one group of neurons represented the planning of instructed vocalizations, a second and largely separate group of neurons encoded the preparation of instructed hand movements. Thus, anatomically intermingled but functionally segregated action preparation networks co-exist in a restricted area of the vlPFC that is the proposed anatomical homolog of Broca's area in the human brain. Hence, clear neural links between the planning of vocal and manual output exist in the nonhuman primate brain prior to the evolution of a human language production system that can access both systems. This suggest a joint evolution of vocal production and hand-action systems in close anatomical vicinity (*Willems and Hagoort, 2007*). The primordial vocal production network, in particular, might have been expanded during primate evolution and put to the service of the speech and language system (*Anderson, 2010*).

## Materials and methods

### Subjects

Two juvenile male macaque monkeys (*Macaca mulatta*), each aged 6 years. and weighing 8.5 and 5.5 kg, respectively, were used for the study. The monkeys worked under a controlled water intake protocol during the experiments and were rewarded with fluid for correct responses. All surgery procedures were accomplished under aseptic conditions under general anesthesia. All procedures have been approved by the responsible national authorities (Regierungspräsidium Tübingen, Germany) under permit ZP 1/15 and comply with German Law and the European Directive 2010/63/EU regulating use of animals in research.

### Behavioral protocol and data acquisition

All recording sessions were carried out in a double-walled, soundproof chamber (IAC Acoustics, Niederkrüchten, Germany). Single-cell recordings were conducted in monkeys trained to perform a computer-controlled 'go/nogo' detection task (*Figure 1*). In alternating trial blocks, the monkey had either to vocalize in response to one of two arbitrary visual cues (red cross or blue square for 'vocal' trials) or to release its hand from a bar in response to another set of visual cues (yellow ring or green square for 'manual' trials). The monkey started a trial by grabbing a bar ('ready'-response). A visual 'nogo'-signal ('pre-cue') appeared for a randomized time period of 1 to 5 s (white square, diameter: 0.5 deg of visual angle) during which the monkey was not allowed to vocalize or release the bar. In 80% of the trials, the 'pre-cue'-signal was followed by a colored visual 'go'-signal (red cross or blue square in the vocalization block; yellow ring or green square in the 'manual' block, all stimuli appeared with equal probability of 40%; diameter: 0.5 deg of visual angle) lasting 3 s. During presentation of the' go'-cue, the monkey had to emit a vocalization or release the bar to receive a liquid reward, respectively. In the vocalization trials, both monkeys produced 'grunt' vocalizations. To control for random responses or timed responses independent from the 'go'-cue, 'catch' trials displaying no 'go'-cue appeared in the remaining 20% of the trials. In 'catch' trials, the 'pre-cue' remained unchanged for 3 s and the monkey had to remain silent in 'vocal' trials (withhold vocal output), or to keep holding the bar in 'manual' trials depending on the block ('catch'-trials).

Correct responses ('vocal' or 'manual', respectively) during 'go'-signals were defined as 'hits'; calls or hand releases during 'catch'-trials counted as 'false alarms' according to the 'go/nogo' detection paradigm. To ensure that monkeys were under stimulus control during each session, we computed d'-sensitivity-values derived from signal detection theory (*Nevin, 1969*) by subtracting z-scores (normal deviates) of median 'hit'-rates from z-scores of median 'false alarm'-rates for each condition separately. Detection threshold for d'-values was set to 1.5 (*Gavrilov et al., 2017*; *Hage and Nieder, 2013*).

Monkeys were head-fixed during the experiment in 57 cm in front of a computer screen and maintained a constant distance of 5 cm between the monkey's head and the microphone. Eye movements were monitored via an IR-eye tracking system (ISCAN, Woburn, MA, USA), sampled at 1 kHz and stored with the Plexon system for subsequent analysis.

Stimulus presentation and behavioral monitoring was automated on PCs running the CORTEX program (NIH) and recorded by a multi-acquisition system (Plexon Inc, Dallas, TX). Vocalizations and bar releases were recorded synchronously with the neuronal data. Vocalizations were sampled at 40 kHz via an A/D converter for post-hoc analysis. Both bar releases and vocalizations were detected automatically. Vocalizations were detected in real-time by a MATLAB program (MathWorks, Natick, MA) that calculated online several temporal and spectral acoustic parameters. Vocal on- and offset times were also detected offline by a MATLAB program to ensure precise timing for data analysis. Reward delivery was automatically initiated 100 ms after call detection, which was typically 500 ms after the call onset; thus, reward was provided no earlier than 600 ms after the call onset. Reward for correct hand releases in correct 'manual' trials was provided 300 ms after the release of the bar was detected.

Because of the difference in task difficulty (monkeys find it much harder to vocalize on command than to release a bar), the monkeys were kept motivated by providing more reward in the' vocal' trials than in the 'manual' trials. The amount of the reward was kept constant in 'vocal' trials, while smaller and larger rewards were randomly provided from trial to trial in 'manual' trials. To ensure

that the monkeys did not simply skip the harder vocalization trials, the block would only switch if the monkey correctly vocalized in 25 trials (delayed and randomized retry of skipped trials).

## Neurophysiological recordings

Extracellular single-cell recordings were performed in the inferior frontal region. The recording well and craniotomy were centered around the inferior arcuate sulcus (AS). This provided access to the ventrolateral PFC (ventral pre-arcuate region, VPA, and regions inside the inferior arcuate sulcus, ASi, at least 6 mm below the cortical surface) and the adjacent rostroventral premotor cortex (PMrv). The recording sites and the position of the recording chambers were localized using stereotaxic reconstructions from individual magnetic resonance images (see *Figure 4*).

Arrays of 4–8 glass-coated tungsten microelectrodes (1 and 2 MΩ, impedance, Alpha Omega, Alpharetta GA) were inserted each recording day using a grid with 1 mm spacing. Neurons were randomly selected; no attempts were made to preselect neurons. Signal acquisition, amplification, filtering and offline spike sorting were carried out using the Plexon system. Data analysis was accomplished using MATLAB (MathWorks). We analyzed all well-isolated neurons with mean discharge rate of more than 1 Hz, recorded during both conditions with at least 7 'hit' trials in each of the condition.

## Neuronal data analysis

### Eye movement- and fixation-correlated neurons

Eye movements were monitored via a computer-controlled IR-eye tracking system (ISCAN), sampled at 1 kHz and stored with the Plexon system for offline analysis. Due to the complex behavioral design, monkeys were not trained to additionally maintain eye position. However, we observed that vocalizations were typically accompanied by large eye movements. Therefore, neuronal activity was analyzed post-hoc to exclude eye movement and eye fixation-related neurons.

To detect saccade-related neuronal activity, we tested whether neuronal activity during the 'precue' period was a function of saccade direction (saccades were defined as eye positions that changed for more than 4 degrees of visual angle within 4 ms). The directional vectors of the saccadic eye movement were produced by comparing the eye position 50 ms prior to saccade onset and the eye position 50 ms after saccade onset. Directional vectors were rearranged into eight groups and peri-event-time histograms were generated. We performed a non-parametric one-way analysis of variance (Kruskal-Wallis test) to test for significant differences in firing rate between vector groups within 200 ms around saccade onset. Neurons that showed saccadic eye-movement-correlated activity were omitted from analysis on vocalization-correlated and hand-movement-correlated activity.

To detect fixation-related neurons, we tested all neurons that did not show saccade-related neuronal activity (neurons with saccade-related activity were already excluded). 'Fixation neurons' are defined as neurons that increase their firing rates after cue fixation. We, therefore, analyzed the neuronal data during recording periods in which a cue was present ('pre-cue', 'go'-signal) and the animals fixated this cue stimulus. In these epochs, we analyzed all fixation period of at least 200 ms in duration. We performed a Wilcoxon sign rank test (p<0.05) to test for significant increases of firing rates during the fixation period (100 ms to 200 ms after fixation onset) compared to firing rates within a 100 ms window before.' Fixation neurons' were omitted from further analysis.

### Motor preparation correlated neurons

We analyzed pre-response activity in a 450 ms window before the onset of either vocalizations or hand movements and compared firing rates (FRs) within this time interval to baseline activity. Baseline activity was derived from a 450 ms window prior to' go'-cue onset. Trials with shorter than 450 ms response latencies (mainly 'manual' trials) were discarded. We performed a Wilcoxon signed rank test (p<0.05) to analyze whether neurons significantly modulated FRs during vocalization and/or 'manual' trials compared to baseline activity.

Additionally, we performed a Mann-Whitney-U test (p<0.05) to test whether neurons showed a significant difference in FRs to one of the two 'go' cues used for each (vocal or manual) response type. Neurons that preferred one of the 'go' stimuli over the other were excluded from further analysis to avoid confounds by sensory coding. Neurons with significant difference in FRs in vocalization

trials were determined as vocalization-correlated neurons. Similarly, neurons with significant difference in FRs in 'manual' trials were determined as hand-correlated neurons.

Neuronal latency was estimated based on peristimulus time histograms and was defined by the first 20 consecutive 50 ms bins that crossed the threshold of three times standard deviation of the baseline firing rate (averaged activity in 450 ms prior to 'go' cue onset). We tested whether the neuronal latency of vocalization- and hand- correlated neurons differed between the areas by a Kruskal-Wallis test.

We did not analyze neuronal activity after response onset, that is when the monkeys vocalized or executed a hand movement. This is because an inseparable mixture of response factors emerged during these responses. For instance, sensory-tactile and motor-execution signals co-occurred during the response period in 'manual' trials, whereas respiratory and orofacial motor commands as well as auditory input resulting from the monkeys' own vocalizations coincided in the response period in 'vocal' trails. Moreover, reward was given immediate after the monkeys' response, suggesting reward expectation and reward delivery as further confounding factors. For all these reasons, and because our task was only designed to differentiate premotor activity, we refrained from analyzing neuronal activity during response execution.

## Volitional and spontaneous responses

Both monkeys sometimes produced calls ('grunt' and 'coo' vocalizations) in between trials during the recording. These calls were uttered without an external cue and therefore defined as spontaneous vocalizations. To compare pre-vocal neuronal activity between volitional calls uttered during cued vocalization trials and spontaneous calls produced in between cued trials, we again analyzed discharges in a 450 ms window before vocal onset. Included in the analysis were only neurons that were recorded while the monkeys produced at least three spontaneous vocalizations. We performed Mann–Whitney U-tests ($p<0.05$) to analyze significant differences in FRs of single neurons between volitional and spontaneous vocalizations.

During the session, the monkeys had to release the bar in order to initiate the trials or to respond to the 'go'-stimulus; otherwise, they had to keep holding the bar throughout the trial. However, sometimes the monkeys released the bar several times in a row (mainly in the vocalization block), which would neither initiate a trial (rather abort it) nor be a response because no 'go'-cue was presented. These responses were defined as spontaneous hand movements. Similarly, to the cued and spontaneous vocalizations, we performed Mann–Whitney U-tests ($p<0.05$) to analyze significant differences in FRs of single neurons between cued and spontaneous hand movements.

## Frequency analysis

To analyze whether cued vocalizations and cued hand movement are encoded by the same population of neurons, we calculated the theoretical chance probability of neurons encoding both response types. To that aim, we multiplied the proportions of vocalization- and hand-correlated neurons recorded in each of the three brain areas (PMrv, ASi, and VPA). We then tested with a binomial test whether the observed probability of neurons encoding vocalizations and hand movements was different from chance expectation.

## Sliding ROC analysis

We determined the coding quality of vocalization- and hand-correlated neurons prior to volitional actions by applying receiver operating characteristic (ROC) analysis derived from Signal Detection Theory. To that aim, we performed a sliding window analysis, using spike rates in overlapping 200 ms windows stepped in 10 ms increments starting from 2000 ms before to 1000 ms after response onset. The area under the ROC curve (AUROC) is a nonparametric measure of the discriminability of two distributions. The AUROC indicates the probability with which an ideal observer can tell apart a meaningful signal from a noisy background. Values of 0.5 indicate no separation, and values of 1 or 0 imply perfect discriminability. The AUROC considers both the difference between distribution means as well as distribution variability and is therefore an unbiased indicator of signal quality. We used AUROC to quantify the quality of response type coding in the pre-response period for vocalization- and hand-correlated neurons separately. We calculated the AUROC for each neuron using the spike rate distributions in vocalization and 'manual' trials.

To test for significance, we used a permutation test. Trials of both conditions were shuffled and randomly assigned to labels 'vocal' and 'manual' 1000 times, in order to create a null distribution of AUROC values around 0.5. A neuron's actual AUROC values were determined to be significant different from 0.5 if they exceeded the highest or lowest 2.5th percentile of this null distribution (p<0.05). The neuronal latency was assigned as the first window of three consecutive windows with AUROC values that significantly differed from 0.5. The latency could not be determined for five vocalization-correlated neurons and two hand-correlated neurons.

### Gaussian-process factor analysis

To study how the recorded neuronal populations as a whole dynamically encode the initiation of volitional call production and 'manual' responses, we represent population activity as trajectories in a multidimensional state space. At each point of time, the activity of recorded neurons on individual trials corresponds to a point in the n-dimensional space with each dimension representing the activity of all pseudo-simultaneously recorded neurons on an individual trial.

We performed a Gaussian-Process-Factor Analysis (GPFA) to extract smooth, low dimensional neuronal trajectories from the high-dimensional noisy spiking activity. The GPFA is a dimensionality reduction method which is applied to single-trial population activity and extracts single trial neuronal trajectories. This method allows the comparison of population activity across trials and characterizes how the population activity progress over time (*Cunningham and Yu, 2014*). In comparison to other dimensionality methods, GPFA combines the smoothing and the dimensionality reduction and performs both operations simultaneously. GPFA was performed using MATLAB toolboxes (*Yu et al., 2009*).

The data were reduced to eight latent dimensions using 20 ms non-overlapping bins. To create a population of pseudo-simultaneously recorded neurons, 17 trials (VPA) or 15 trials (ASi and PMrv) were randomly selected from all available trials (for each condition separately). Based on this criterion, we included 180 neurons recorded in VPA, 81 neurons and 36 neurons in ASi and PMrv, respectively. Using GPFA, we calculated two average state-space trajectories for each condition, each across one half all single-trial trajectories defined by the top three dimensions (that explain 58% of the neuronal covariance) of orthonormalized trajectories. The Euclidean Distance between the two trajectories determined for trials of the same condition ('vocal' or 'manual') was used as a measure of 'within-response coding'. As a measure for 'cross-response coding', we calculated the Euclidean Distance between the four trajectories determined for vocalization trials and 'manual' trials. The distance within a time window of 100–600 ms before the corresponding 'go'-cue onset was used as baseline and was subtracted from distances calculated for each time point. The entire procedure of selecting trials and performing the GPFA analysis was repeated 100 times to account for differences in selecting the data.

## Acknowledgements

We thank Diana Liao for reading an earlier version of the manuscript.

## Additional information

### Funding

| Funder | Grant reference number | Author |
| --- | --- | --- |
| Universtity of Tuebingen | Intramural grant | Andreas Nieder |

The funders had no role in study design, data collection and interpretation, or the decision to submit the work for publication.

### Author contributions

Natalja Gavrilov, Resources, Software, Formal analysis, Validation, Investigation, Visualization, Methodology, Writing - original draft, Writing - review and editing; Andreas Nieder, Conceptualization, Resources, Data curation, Supervision, Funding acquisition, Validation, Methodology, Writing - original draft, Project administration, Writing - review and editing

## Author ORCIDs
Andreas Nieder (iD) https://orcid.org/0000-0001-6381-0375

## Ethics
Animal experimentation: All procedures have been approved by the responsible national authorities (Regierungspräsidium Tübingen, Germany) under permit ZP 1/15 and comply with German Law and the European Directive 2010/63/EU regulating use of animals in research.

## Decision letter and Author response
Decision letter https://doi.org/10.7554/eLife.62797.sa1
Author response https://doi.org/10.7554/eLife.62797.sa2

# Additional files

## Supplementary files
• Transparent reporting form

## Data availability
Matlab code and data to reproduce Figures 2, 3, 5, 7-9 are publicly available at https://doi.org/10.17632/3w3whp7r44.1.

The following dataset was generated:

| Author(s) | Year | Dataset title | Dataset URL | Database and Identifier |
|---|---|---|---|---|
| Nieder A | 2020 | Gavrilov, Nieder_Distinct neural networks for the volitional control of vocal and manual actions in the monkey homologue of Broca's area | http://dx.doi.org/10.17632/3w3whp7r44.1 | Mendeley Data, 10.17632/3w3whp7r44.1 |

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
