## [Decision Letter]

**Acceptance summary:**

This article examined neuronal responses in the homologues of Broca's region in the macaque monkey, i.e. areas 44 and 45. Single neurons recorded from areas 44, 45, and ventral premotor area 6 predominantly signalled the impending vocal or, to a lesser extent, manual action, but not both. These results suggest that the nonhuman primate inferior frontal cortex controls the initiation of volitional utterances via a dedicated network of vocally selective neurons that may be the putative evolutionary precursor of Broca's area in the human brain - a region involved in the control of speech production. This paper demonstrates potentially important findings about the potential homologue of human vocal motor cortex.

**Decision letter after peer review:**

Thank you for submitting your article "Distinct neural networks for the volitional control of vocal and manual actions in the monkey homologue of Broca's area" for consideration by *eLife*. Your article has been reviewed by Joshua Gold as the Senior Editor, a Reviewing Editor, and three reviewers. The following individual involved in review of your submission has agreed to reveal their identity: Steve Eliades (Reviewer #3).

The reviewers have discussed the reviews with one another and the Reviewing Editor has drafted this decision to help you prepare a revised submission.

We would like to draw your attention to changes in our revision policy that we have made in response to COVID-19 (https://elifesciences.org/articles/57162). Specifically, we are asking editors to accept without delay manuscripts, like yours, that they judge can stand as *eLife* papers without additional data, even if they feel that they would make the manuscript stronger.

Summary:

This article examined neuronal responses in the homologues of Broca's region in the macaque monkey, i.e. areas 44 and 45. The neurons signaled the volitional initiation of vocalizations. Single neurons recorded from areas 44, 45, and ventral premotor area 6 predominantly signaled the impending vocal or, to a lesser extent, manual action, but not both. These results suggest that the nonhuman primate inferior frontal cortex controls the initiation of volitional utterances via a dedicated network of vocally-selective neurons that may be the putative evolutionary precursor of Broca's area in the human brain that controls speech production. All three reviewers agree that the paper demonstrates potentially important findings about the potential homolog of human vocal motor cortex. It follows from previous work by this group, investigating the potential homology of the nonhuman primate ventral frontal lobe and vocal behavior.

Essential revisions:

1) The authors do not report or comment on any differences in the timing of activity in areas 44 and 45. If area 44 controls directly premotor area 6, but area 45 is a higher cognitive area that controls area 45, it is possible that neurons in area 45 may have reaction times that are longer than those in area 44. Also, were there any differences in timing of the ventral area 6 vocal neurons?

2) It is interesting that the superior longitudinal fasciculus, branch III (SLF III) links rostral inferior parietal lobule with areas 6 and 44, while the superior longitudinal fasciculus, branch II (SLF II) links area 45 with posterior inferior parietal lobule (i.e. angular gyrus) (see Petrides and Pandya, 1984 and also Petrides, 2014. Barbeau, Descoteaux, and Petrides, 2020.

3) The use of the term “volitional vocalization” is confusing. One would expect that both the cued vocalizations and the un-cued "spontaneous" vocalizations are volitional to some degree. The term "cued" is much clearer and accurately portrays the nature of these vocal responses without anthropomorphic interpretations.

4) While the tremendous effort to record spontaneous vocalizations is greatly appreciated it does not seem as if the data arguing that the cued vocalizations are different from spontaneous vocalizations is strong enough. There appear to be very few spontaneous calls and it is not clear that the sample size is large enough for a valid comparison.

5) The data are interpreted in terms of the cytoarchitectonic areas of the brain in the ventral prefrontal and premotor areas (45, 44 and 6v). However, the authors do not offer any cytoarchitectonic evidence to justify use of these designations since, presumably, their subjects are still involved in research. In most physiological studies where this is the case, functional areal designations are used such as ventral premotor area (area 6) and ventral pre-arcuate cortex for the cortical region that includes areas 45 and anterior 44 (Bichot et al., 2015). Without histological verification the authors should use the functional designations since they cannot confirm the precise location of the cells they are recording and this may prove to be important in the future.

6) How separable are the hand and vocal responses, really? For example, many of the examples and population averages in Figure 5 show hand-like peaks round 0 ms in vocalization-correlated neurons. While these may reflect some of the overlapping neurons discussed in Figure 6, I am unsure. Alternatively, were these due to the rate of incorrect hand releases? An analysis based on incorrect responses might be useful to distinguish.

7) Related to 6), the populations averages in Figure 5 also show that these vocal or hand neurons often respond during the "other" task, just in a more delayed fashion. Any explanation for this pattern? It would also be useful to know at what time points these responses statistically diverged (both from baseline and from each other). The authors might consider a time point-by-point statistical test for the population averages, with some sort of correction for multiple comparisons.

8) As to interpretation of Figure 7, on the difference between cued and spontaneous activities: I believe this issue has been raised in the authors' past work, about the interpretation of the lack of neural responses during non-cued calls, and whether the premotor activity really reflects the training or the natural vocal control. The data presented here raise similar issues -- A similar pattern is seen here for the hand movements. It’s harder to argue that the non-cued hand movements were truly spontaneous or non-volitional, as has been argued for the calls, unless all of them were random errors. I think some more discussion on this point would be useful.

9) Regarding the different reward conditions, with higher reward for vocalizations and random reward for hand movements: Can one be sure that reward expectation is not playing a role in response differences? Perhaps not, particularly with the different temporal patterns of responses, but this is not clear.

[Editors' note: further revisions were suggested prior to acceptance, as described below.]

Thank you for submitting your article "Distinct neural networks for the volitional control of vocal and manual actions in the monkey homologue of Broca's area" for consideration by *eLife*. Your article has been reviewed by Joshua Gold as the Senior Editor, a Reviewing Editor, and two reviewers. The following individual involved in review of your submission has agreed to reveal their identity: Steve Eliades (Reviewer #3).

The reviewers have discussed the reviews with one another and the Reviewing Editor has drafted this decision to help you prepare a revised submission.

Summary:

This article examined neuronal responses in the homologues of Broca's region in the macaque monkey, i.e. areas 44 and 45. The neurons signaled the volitional initiation of vocalizations. Single neurons recorded from areas 44, 45, and ventral premotor area 6 predominantly signaled the impending vocal or, to a lesser extent, manual action, but not both. These results suggest that the nonhuman primate inferior frontal cortex controls the initiation of volitional utterances via a dedicated network of vocally-selective neurons that may be the putative evolutionary precursor of Broca's area in the human brain that controls speech production. All three reviewers agree that the paper demonstrates potentially important findings about the potential homolog of human vocal motor cortex. It follows from previous work by this group, investigating the potential homology of the nonhuman primate ventral frontal lobe and vocal behavior.

Essential revisions:

The revised text has addressed the main issues raised in the review. However, there are several errors in the phrases of the manuscript and the authors should do a thorough check and revision for grammar and phrasing.

There are many inconsistencies in the style of the references. The authors should examine the required style of the journal and follow it, and thoroughly check all of the references. For example, in most but not all references, the volume number and page numbers are provided. In some references only the volume is provided but not the page numbers. Also, in some references the title is written with capital letters for the first letter of the words (as an example, see Bichot et al.,), but in most other references the words are not written with capital first letters. Also, in some references, the volume is provided in a strange manner (see Balter as an example).

There is still a concern related to point 7 from the original decision letter, which the authors have not adequately addressed. (The concern may not previously have been clear to the authors) Specific timing analyses are hard to perform. The task-specific labels are based, understandably, on the pre-vocal response period. However, the average activities of these vocal/hand-specific also show activity for the other task during the later time period once the motor task has begun. This would require repeating the comparisons during the motor period, and not just the pre-motor period. This would not be captured in Figure 6 which appears to be a comparison only of the pre-motor activity. How often do task-specific neurons show responses to the other task during this later time period, and how do we interpret this?

---

## [Author Response]

Essential revisions:1) The authors do not report or comment on any differences in the timing of activity in areas 44 and 45. If area 44 controls directly premotor area 6, but area 45 is a higher cognitive area that controls area 45, it is possible that neurons in area 45 may have reaction times that are longer than those in area 44. Also, were there any differences in timing of the ventral area 6 vocal neurons?

We thank the reviewer for this valuable suggestion. We now analyzed the neuronal latencies for vocalization- and hand-correlated neurons (for each area). We report that we did not find significant differences subsection “Response type correlated neurons”. The analysis is explained in the Materials and methods.

2) It is interesting that the superior longitudinal fasciculus, branch III (SLF III) links rostral inferior parietal lobule with areas 6 and 44, while the superior longitudinal fasciculus, branch II (SLF II) links area 45 with posterior inferior parietal lobule (i.e. angular gyrus) (see Petrides and Pandya, 1984 and also Petrides, 2014. Barbeau, Descoteaux, and Petrides, 2020.

We are now discussing this interesting anatomical segregation in the Discussion.

3) The use of the term “volitional vocalization” is confusing. One would expect that both the cued vocalizations and the un-cued "spontaneous" vocalizations are volitional to some degree. The term "cued" is much clearer and accurately portrays the nature of these vocal responses without anthropomorphic interpretations.

We agree with the reviewer that it is good practice to describe observations without interpretation in the Results section. We therefore now exclusively speak of “cued” versus “spontaneous” vocalizations/hand movements throughout the result section and in all figures. In the discussion, when we interpret our findings, we do resort to the term “volitional” to interpret our findings. In order to provide a clear operational definition of the term “volitional”, we inserted a list of three criteria in the Introduction that have to be fulfilled in unison. This list of criteria is derived from tests in clinical neurology, and we provide examples. We therefore think it is justified to use this term. We don’t think ascribing “volitional behavior” in this context is an anthropomorphic interpretation; animal certainly have cognitive control over some of their behaviors and thus show “volitional” control. Our current and past data provides objective neurophysiological evidence that the initiation of vocalizations relies on two categorically distinct processes that we need to name and interpret. Also, please note that we and others have used this terminology in many publications before (e.g. Hage and Nieder, 2013; Gavrilov et al., 2017; Brecht et al., 2019), indicating that it is of value in the field of the study of vocalizations.

4) While the tremendous effort to record spontaneous vocalizations is greatly appreciated it does not seem as if the data arguing that the cued vocalizations are different from spontaneous vocalizations is strong enough. There appear to be very few spontaneous calls and it is not clear that the sample size is large enough for a valid comparison.

We apologize for causing a misunderstanding. Indeed, we found a statistically significant difference when we compared activity to cued versus spontaneous vocalizations in the population of 14 neurons for which enough spontaneous vocalizations could be recorded (Wilcoxon signed rank test, p<0.01). This argues that the coding of cued vocalizations is different from the coding of spontaneous vocalizations. We clarified this in subsection “Cued versus spontaneous responses”.

5) The data are interpreted in terms of the cytoarchitectonic areas of the brain in the ventral prefrontal and premotor areas (45, 44 and 6v). However, the authors do not offer any cytoarchitectonic evidence to justify use of these designations since, presumably, their subjects are still involved in research. In most physiological studies where this is the case, functional areal designations are used such as ventral premotor area (area 6) and ventral pre-arcuate cortex for the cortical region that includes areas 45 and anterior 44 (Bichot et al., 2015). Without histological verification the authors should use the functional designations since they cannot confirm the precise location of the cells they are recording and this may prove to be important in the future.

As suggested by the reviewer, we now use anatomically-descriptive terms to differentiate the three recorded frontal areas. The conversion of the Brodmann-areas is as follows:

Area 6 -> rostral-ventral premotor cortex (PMrv)

Area 44 -> inferior arcuate sulcus (ASi)

Area 45 -> ventral pre-arcuate region (VPA)

We replaced our original Brodmann-area definitions with the new term throughout the manuscript, for instance, Introduction, Results, Materials and methods.

6) How separable are the hand and vocal responses, really? For example, many of the examples and population averages in Figure 5 show hand-like peaks round 0 ms in vocalization-correlated neurons. While these may reflect some of the overlapping neurons discussed in Figure 6, I am unsure. Alternatively, were these due to the rate of incorrect hand releases? An analysis based on incorrect responses might be useful to distinguish.

Indeed, some of the example neurons we show in Figure 5 are from the pool of overlapping neurons we discuss in Figure 6. This is now clarified (subsection “Response type correlated neurons”). All the responses shown in Figure 5 are exclusively from correct trials, which is now also clearly stated.

7) Related to 6), the populations averages in Figure 5 also show that these vocal or hand neurons often respond during the "other" task, just in a more delayed fashion. Any explanation for this pattern? It would also be useful to know at what time points these responses statistically diverged (both from baseline and from each other). The authors might consider a time point-by-point statistical test for the population averages, with some sort of correction for multiple comparisons.

Yes, a fraction of the neurons responded to both types of movements. This finding is analyzed in subsection “Response type correlated neurons” and broken down in Figure 6. We appreciate the reviewer’s suggestion to consider a time point-by-point statistical test for the population averages in order to explore the time points of response divergence. Unfortunately, this is not possible because of the radically different reaction times of the monkeys: hand movements were executed much faster after the go-cue than vocalizations, and vocalization-correlated neurons showed a long-lasting and gradual increase in activity which would reach long into the pre-go-cue period for hand-correlated neurons. This is also demonstrated by our new analysis of the neurons’ response latencies to both response types (subsection “Response type correlated neurons”).

8) As to interpretation of Figure 7, on the difference between cued and spontaneous activities: I believe this issue has been raised in the authors' past work, about the interpretation of the lack of neural responses during non-cued calls, and whether the premotor activity really reflects the training or the natural vocal control. The data presented here raise similar issues -- A similar pattern is seen here for the hand movements. It’s harder to argue that the non-cued hand movements were truly spontaneous or non-volitional, as has been argued for the calls, unless all of them were random errors. I think some more discussion on this point would be useful.

To address this issue of “volitional” versus “spontaneous”, we inserted a clear operational definition of these response types in the Introduction. The comparison of cued versus spontaneous vocalizations and hand movements shown in Figure 7 shows impressive and highly significant activity differences (statistics is provided in subsection “Cued versus spontaneous responses”). This strongly argues that the coding of cued vocalizations is different from the coding of spontaneous vocalizations. We are now evaluating this important point in the Discussion.

9) Regarding the different reward conditions, with higher reward for vocalizations and random reward for hand movements: Can one be sure that reward expectation is not playing a role in response differences? Perhaps not, particularly with the different temporal patterns of responses, but this is not clear.

This is a valid point and we are discussing this aspect now in the Discussion. We explain why higher reward for “vocal” trials was necessary, and why it is unlikely that differences in reward expectation might explain the observed coding differences

[Editors' note: further revisions were suggested prior to acceptance, as described below.]

Essential revisions:The revised text has addressed the main issues raised in the review. However, there are several errors in the phrases of the manuscript and the authors should do a thorough check and revision for grammar and phrasing.

We thoroughly checked the manuscript and corrected grammar and phrasing errors.

There are many inconsistencies in the style of the references. The authors should examine the required style of the journal and follow it, and thoroughly check all of the references. For example, in most but not all references, the volume number and page numbers are provided. In some references only the volume is provided but not the page numbers. Also, in some references the title is written with capital letters for the first letter of the words (as an example, see Bichot et al.,), but in most other references the words are not written with capital first letters. Also, in some references, the volume is provided in a strange manner (see Balter as an example).

We corrected the style of the references and made sure they concur with the requirements of *eLife*.

There is still a concern related to point 7 from the original decision letter, which the authors have not adequately addressed. (The concern may not previously have been clear to the authors) Specific timing analyses are hard to perform. The task-specific labels are based, understandably, on the pre-vocal response period. However, the average activities of these vocal/hand-specific also show activity for the other task during the later time period once the motor task has begun. This would require repeating the comparisons during the motor period, and not just the pre-motor period. This would not be captured in Figure 6 which appears to be a comparison only of the pre-motor activity. How often do task-specific neurons show responses to the other task during this later time period, and how do we interpret this?

We agree that it would be interesting to know more about activity during the time period once the motor task had begun. However, our task design is not suited to analyze such signals. The goal of this study was solely to identify neuronal activity that is correlated with the instructed initiation of vocalizations and/or hand movements. Our task was entirely geared toward premotor activity for both “manual” and “vocal” trials.

We did not analyze neuronal activity after response onset, i.e. when the monkeys vocalized or executed a hand movement. This is because an inseparable mixture of response factors emerged during these responses. For instance, sensory-tactile and motor-execution signals co-occurred during the response period in “manual” trials, whereas respiratory and orofacial motor commands as well as auditory input resulting from the monkeys’ own vocalizations coincided in the response period in “vocal” trails. Moreover, reward was given immediate after the monkeys’ response, suggesting reward expectation and reward delivery as further confounding factors. For all these reasons, and because our task was only designed to differentiate premotor activity, we prefer to refrain from analyzing neuronal activity during response execution. This argumentation is now provided in subsection “Motor preparation correlated neurons”.